# Ω-Loop mutations control dynamics of the active site by modulating the hydrogen-bonding network in PDC-3 β-lactamase

**Shuang Chen[1], Andrew R Mack[2,3], Andrea M Hujer[2,4], Christopher R Bethel[2], Robert A Bonomo[2,3,4,5,6,7], Shozeb Haider[1,8,9]\***

[1]University College London, London, United Kingdom; [2]Research Service, Louis Stokes Cleveland, Department of Veterans Affairs Medical Center, Cleveland, United States; [3]Department of Molecular Biology and Microbiology, Case Western Reserve University School of Medicine, Cleveland, United States; [4]Department of Medicine, Case Western Reserve University School of Medicine, Cleveland, United States; [5]Clinician Scientist Investigator, Louis Stokes Cleveland Department of Veterans Affairs Medical Center, Cleveland, United States; [6]Departments of Pharmacology, Biochemistry, and Proteomics and Bioinformatics, Case Western Reserve University School of Medicine, Cleveland, United States; [7]CWRU-Cleveland VAMC Center for Antimicrobial Resistance and Epidemiology (Case VA CARES) Cleveland, Cleveland, United States; [8]UCL Centre for Advanced Research Computing, London, United Kingdom; [9]University of Tabuk (PFSCBR), Tabuk, Saudi Arabia

**\*For correspondence:**
shozeb.haider@ucl.ac.uk

## eLife Assessment

This article uses adaptive-bandit simulations to describe the dynamics of the *Pseudomonas*-derived chephalosporinase PDC-3 β-lactamase and its mutants to better understand antibiotic resistance. The finding that clinically observed mutations alter the flexibility of the Ω- and R2-loops, reshaping the cavity of the active site, is **valuable** to the field. The evidence is considered **incomplete**, however, with the need for analysis to demonstrate equilibrium weighting of adaptive trajectories and related measures of statistical significance.

**Abstract** The expression of antibiotic-inactivating enzymes, such as *Pseudomonas*-derived cephalosporinase-3 (PDC-3), is a major mechanism of intrinsic resistance in bacteria. Using reinforcement learning-driven molecular dynamics simulations and constant pH MD, we investigate how clinically observed mutations in the Ω-loop (at residues V211, G214, E219, and Y221) alter the structure and function of PDC-3. Our findings reveal that these substitutions modulate the dynamic flexibility of the Ω-loop and the R2-loop, reshaping the cavity of the active site. In particular, E219K and Y221A disrupt the tridentate hydrogen bond network around K67, thus lowering its *pKa* and promoting proton transfer to the catalytic residue S64. Markov state models reveal that E219K achieves enhanced catalysis by adopting stable, long-lived 'active' conformations, whereas Y221A facilitates activity by rapidly toggling between bond-formed and bond-broken states. In addition, substitutions influence key hydrogen bonds that control the opening and closure of the active-site pocket, consequently influencing the overall size. The pocket expands in all nine clinically identified variants, creating additional space to accommodate bulkier R1 and R2 cephalosporin side chains. Taken together, these results provide a mechanistic basis for how single residue substitutions in the Ω-loop affect catalytic activity. Insights into the structural dynamics of the catalytic site advance our

understanding of emerging $\beta$-lactamase variants and can inform the rational design of novel inhibitors to combat drug-resistant *P. aeruginosa*.

## Introduction

*Pseudomonas aeruginosa* is a ubiquitous Gram-negative bacterium from the family Pseudomonadaceae (*Pang et al., 2019*). This pathogen is commonly found in hospitals and other healthcare settings, where it can cause infections in people who are immunocompromised or have chronic conditions (*Kerr and Snelling, 2009*). Pseudomonal infections are associated with high morbidity and mortality in many groups, including patients with cystic fibrosis, pneumonia, and chronic obstructive pulmonary disease (*Curran et al., 2018*; *Jurado-Martín et al., 2021*; *Malhotra et al., 2019*; *Reynolds and Kollef, 2021*). $\beta$-Lactam antibiotics, characterized by the presence of a $\beta$-lactam ring in their chemical structure, are often the first-line treatment for bacterial infections as they tend to have fewer side effects and are less toxic than other antibiotics (*Mora-Ochomogo and Lohans, 2021*). Mechanistically, $\beta$-lactams work by inhibiting the synthesis of the bacterial cell wall, which is necessary for the survival and growth of bacteria (*Lima et al., 2020*). $\beta$-lactam antibiotics are usually effective against a wide range of bacteria, but can lose their effectiveness if the bacteria develop resistance to them. *P. aeruginosa* is known for its ability to develop resistance to multiple classes of antimicrobial drugs (*Horcajada et al., 2019*; *Pang et al., 2019*; *Spagnolo et al., 2021*). Therefore, the greatest challenges to eradicating *P. aeruginosa* infections are multidrug-resistant and extensively drug-resistant isolates. The World Health Organization has identified *P. aeruginosa* as a top-priority pathogen for research and

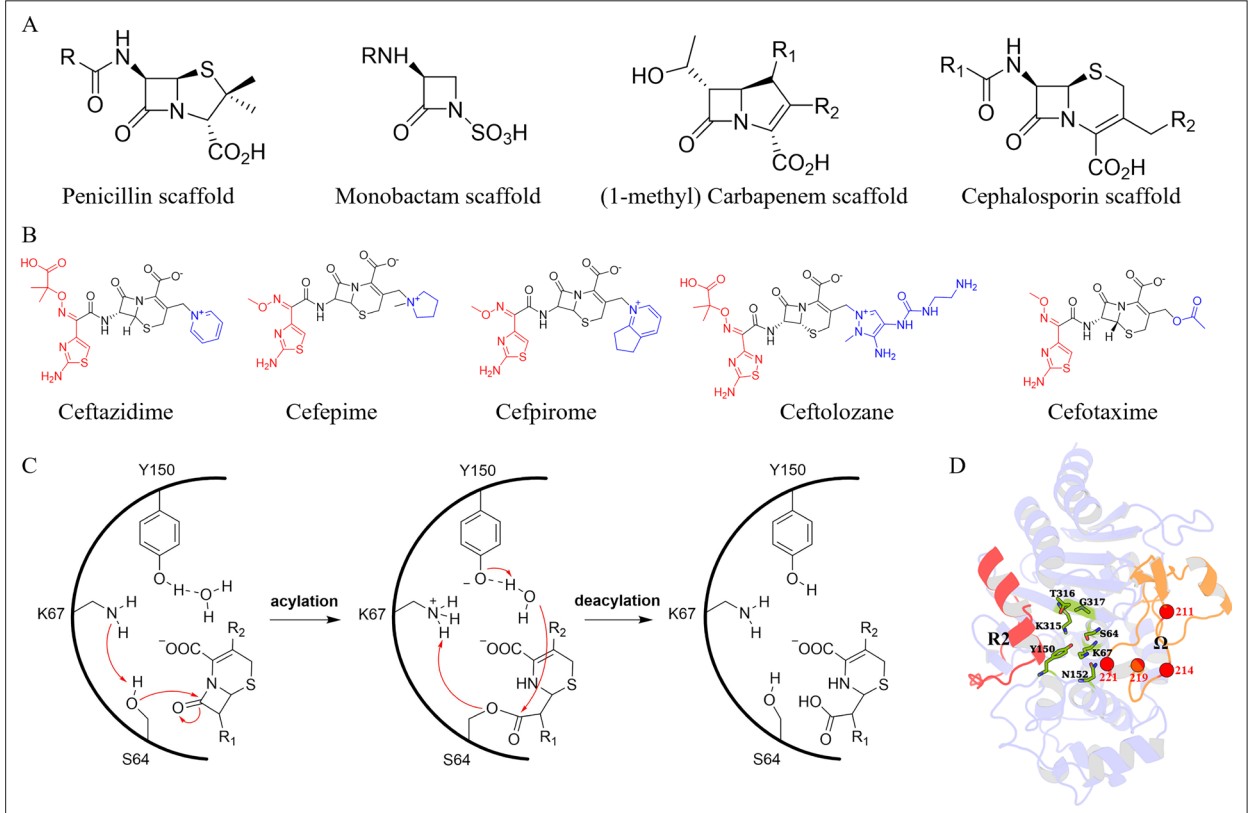

**Figure 1.** Structures and catalytic mechanism of $\beta$-lactam antibiotics and PDC-3 $\beta$-lactamase. (**A**) Structures of representative $\beta$-lactam antibiotics. The notation R (R1/R2) represents the point of addition of functional groups. (**B**) Structures of commonly used cephalosporins. The R1 side chains of the antibiotics are shown in red, while the R2 side chains are marked in blue. (**C**) General mechanism of PDC-3 $\beta$-lactamase hydrolysis of cephalosporins. (**D**) The overall structure of the protein is shown in a cartoon representation (PDB ID: 4HEF). The $\Omega$-loop and R2-loop are colored orange and red, respectively. The conserved residues in the active site are colored green and highlighted as sticks. The positions of all mutations (V211A/G, G214A/R, E219A/G/K, and Y221A/H) are highlighted as red spheres.

development of new antibiotics due to its ability to cause serious infections and its increasing resistance to currently available treatment options (*Tacconelli et al., 2018*).

The production of antibiotic-inactivating enzymes is one of the major mechanisms of intrinsic resistance in bacteria (*Munita and Arias, 2016*). *P. aeruginosa* can express a class C $\beta$-lactamase, named *Pseudomonas*-derived cephalosporinase (PDC), which is an antibiotic-inactivating enzyme (*Colque et al., 2022*). PDC-3 is a serine $\beta$-lactamase that can inactivate a broad range of $\beta$-lactam antibiotics, including penicillins, cephalosporins, monobactams, and carbapenems, by breaking the amide bond of the $\beta$-lactam ring through a catalytic serine (*Figure 1A and B*; *Pang et al., 2019*; *Tripathi and Nair, 2013*; *Tripathi and Nair, 2016*). Cephalosporins are known to be highly susceptible to PDC-3 inactivation (*Barnes et al., 2018*). The active site of PDC-3 is located at the intersection of the enzyme's α-helical and α/β domains (*Figure 1D*). The active site can be further divided into two distinct regions: the R1 site and the R2 site. These regions are defined by the specific binding interactions they facilitate with the R1 and R2 side chains of the cephalosporins, respectively (*Figure 1A*). The R1 site is surrounded by the Ω-loop, while the R2 site is encased by the R2-loop, which comprises the α helix H-10. The Ω-loop and the R2-loop are located at opposite ends of the active site, with the catalytic serine residue positioned in the middle (*Jacoby, 2009*). However, the Ω-loop and R2-loop are particularly prone to amino acid substitutions, insertions, and deletions that expand the active site and accommodate larger R1 and R2 groups of the cephalosporins (*Nordmann and Mammeri, 2007*). These modifications have been observed to enhance the capacity of the enzyme to hydrolyze a wider range of $\beta$-lactam antibiotics (*Barnes et al., 2018*). The evolution of PDC-3 $\beta$-lactamase and its amino acid variants, which often result in enhanced catalytic activity and an expanded spectrum of cephalosporin hydrolysis, has garnered considerable interest in scientific research (*Ruedas-López et al., 2022*). As previously reported, several PDC-3 Ω-loop variants, including V211A, V211G, G214A, G214R, E219A, E219G, E219K, Y221A, and Y221H were found in highly drug-resistant *P. aeruginosa* clinical isolates (*Barnes et al., 2018*). All residues in this study are annotated based on the structural alignment-based numbering of class C $\beta$-lactamase scheme (SANC) (*Mack et al., 2020*).

Molecular dynamics (MD) simulations provide valuable insights into the time-evolving dynamic behavior of biomolecules such as proteins (*Hollingsworth and Dror, 2018*). Among various enhanced sampling methods, AdaptiveBandit molecular dynamics (AB-MD) stands out as a powerful, reinforcement learning (RL)-based adaptive sampling strategy (*Pérez et al., 2020*). Compared to a single long equilibrium MD simulation (which can become trapped in a metastable state) or bias-based enhanced sampling techniques like accelerated MD (aMD) and Gaussian accelerated MD (GaMD) that add boost potentials to smooth the energy landscape and facilitate barrier crossing, AB-MD employs a reinforcement learning-inspired multi-armed bandit to adaptively guide multiple short, unbiased trajectories toward regions of underexplored conformational space (*Bhattarai and Miao, 2018*). In a multi-armed bandit, each arm yields a reward, and an agent must balance exploration (trying less-sampled options) and exploitation (focusing on the best-known options). AB-MD applies this idea to MD by treating different regions of conformational space (or states) as the bandit arms. At each iteration, the algorithm decides from which state to launch a new MD simulation, aiming to maximize sampling efficiency while still obtaining an accurate, unbiased representation of the system's equilibrium behavior.

In this study, the AB-MD approach was employed to explore the conformational landscape of PDC-3 and its variants at the atomistic level. Additionally, constant pH MD simulations were performed to examine the protonation-state behavior of key residues in the catalytic site. By analyzing the conformational ensembles and kinetics of PDC-3 through these simulations, we aimed to uncover underlying mechanisms governing the function of PDC-3 and its variants. Understanding the molecular mechanisms underlying PDC-3 function and the development of resistance is of paramount importance in combating *P. aeruginosa* infections. This knowledge can aid in the development of more effective treatments to combat these bacteria.

## Results and discussion

### Amino acid substitutions change the flexibility of Ω-loops and R2-loops

To investigate how the mutations in the Ω-loop affect PDC-3 dynamics, adaptive-bandit molecular dynamics (AB-MD) simulations were carried out for each system. 100 trajectories of 300 ns each (totaling 30 μs per system) were run. Both root-mean-square deviation (RMSD) and root-mean-square

fluctuation (RMSF) analyses provide insights into the dynamic behavior and structural differences of biomolecules (*Maier et al., 2015*; *Prabantu et al., 2022*). Because AB-MD adaptively seeds new unbiased trajectories to expand conformational sampling, RMSD and RMSF are used here to summarize the structural variability and per-residue mobility observed across the collected trajectories. Firstly, the structural variability of the overall conformation of wild-type PDC-3 and its variants over the collected trajectories was investigated using pairwise RMSD analysis. The results of the RMSD analysis reveal that the wild-type PDC-3 displays a relatively low degree of structural variability, indicating that the protein maintains a relatively consistent overall conformation over the collected trajectories (*Figure 2A*, *Figure 2—figure supplement 1*). Similarly, the V211G, G214A, and Y221H variants also exhibit low RMSD values, which suggests that these structures are less flexible. In contrast, the V211A and E219G variants exhibit the highest RMSD values among the set of structures, indicating a high level of structural variability. This implies that these substitutions lead to increased conformational fluctuations over the collected trajectories. The G214R, E219A, E219K, and Y221A variants exhibit RMSD values that are intermediate between the wild-type and the most flexible variants, indicating that these amino acid substitutions have a moderate effect on the structural stability of the protein's conformation, that is, not as significant as the V211A and E219G substitutions.

To identify regions that contribute the most to the conformational changes in the wild-type PDC-3 and its variants, the RMSF values of Cα atoms were calculated. High RMSF values indicate a high degree of flexibility or mobility for the corresponding atoms, while low RMSF values indicate a significant degree of rigidity (*Bornot et al., 2011*). The wild-type PDC-3 and the G214A, G214R, E219G, and Y221A variants exhibit high flexibility in their Ω-loop, as evidenced by the relatively large per-residue RMSF values observed (approximately 4 Å). In contrast, the V211A, V211G, E219K, and Y221H variants display more constrained conformations in the Ω-loop. Notably, the V211A and V211G variants demonstrate the highest stability in the Ω-loop, with average RMSF values around 1.5 Å, whereas the E219K and Y221H variants exhibit intermediate flexibility, with RMSF values averaging between 2 and 2.5 Å. In terms of the R2-loop, wild-type PDC-3 displays a relatively low degree of flexibility while all variants exhibit an increase in structural flexibility. This suggests that these substitutions have a significant impact on the stability of the R2-loop, potentially affecting enzyme function. A detailed observation of the individual variants reveals that the V211A variant exhibits a particularly high degree of flexibility, as evidenced by the comparatively higher RMSF values, followed by the E219G variant. On the other hand, the Y221A and Y221H variants exhibit a relatively lower degree of flexibility, as inferred by the lower RMSF values observed (*Figure 2*). Therefore, the flexibility of PDC-3 is predominantly localized to the Ω- and R2-loops, whereas the remainder of the structure is comparatively rigid.

The importance of Ω-loop and R2-loop in class C β-lactamases has been previously confirmed (*Philippon et al., 2022*). These loops play a crucial role in the binding and activity of the class C β-lactamases (*Philippon et al., 2022*). Specifically, residues V211 and Y221 within the Ω-loop have been identified to engage in hydrophobic interactions with the R1 side chains of cephalosporins. The substitution of V211A has been reported to be associated with acquired resistance to cefepime or cefpirome (*Rodríguez-Martínez et al., 2010*). Additionally, the characteristic aminothiazole ring found in most third-generation cephalosporins interacts with Y221 in an edge-to-face manner, which represents typical quadrupole-quadrupole interactions. However, Y221 can sometimes create steric clashes that prevent ligands from entering the binding sites (*Barnes et al., 2018*; *Powers and Shoichet, 2002*). Deletion of Y221 has been observed to broaden substrate specificity and confer resistance to ceftazidime-avibactam (*Lahiri et al., 2015*). Moreover, the expanded Ω-loop of P99, another member of class C β-lactamases, exhibits conformational flexibility that may facilitate the hydrolysis of oxyimino β-lactams by making the acyl intermediate more accessible to attack by water (*Crichlow et al., 1999*). In terms of the R2-loop, it has been observed that the N289 (N287 in PDC-3) forms hydrogen-bonding interactions with the C4 carboxylate directly in the AmpC/13 (moxalactam) complex (*Crichlow et al., 2001*). Furthermore, the T289, A292, and L293 residues within the R2-loop of class C β-lactamases have been found to exhibit hydrophobic contacts with the dimethyl group in the R2 chains of cephalosporins (*Powers and Shoichet, 2002*). Additional research suggests that the removal of the R2 group in cephalosporins occurs, while the R1 group remains intact (*Chaudhry et al., 2019*; *Perez-Inestrosa et al., 2005*). This observation indicates that the high flexibility of the R2-loop could be a crucial factor in stabilizing substrates during both the acylation and deacylation steps simultaneously. Overall, the flexibility or mobility of the Ω-loops and R2-loops allows the PDC-3

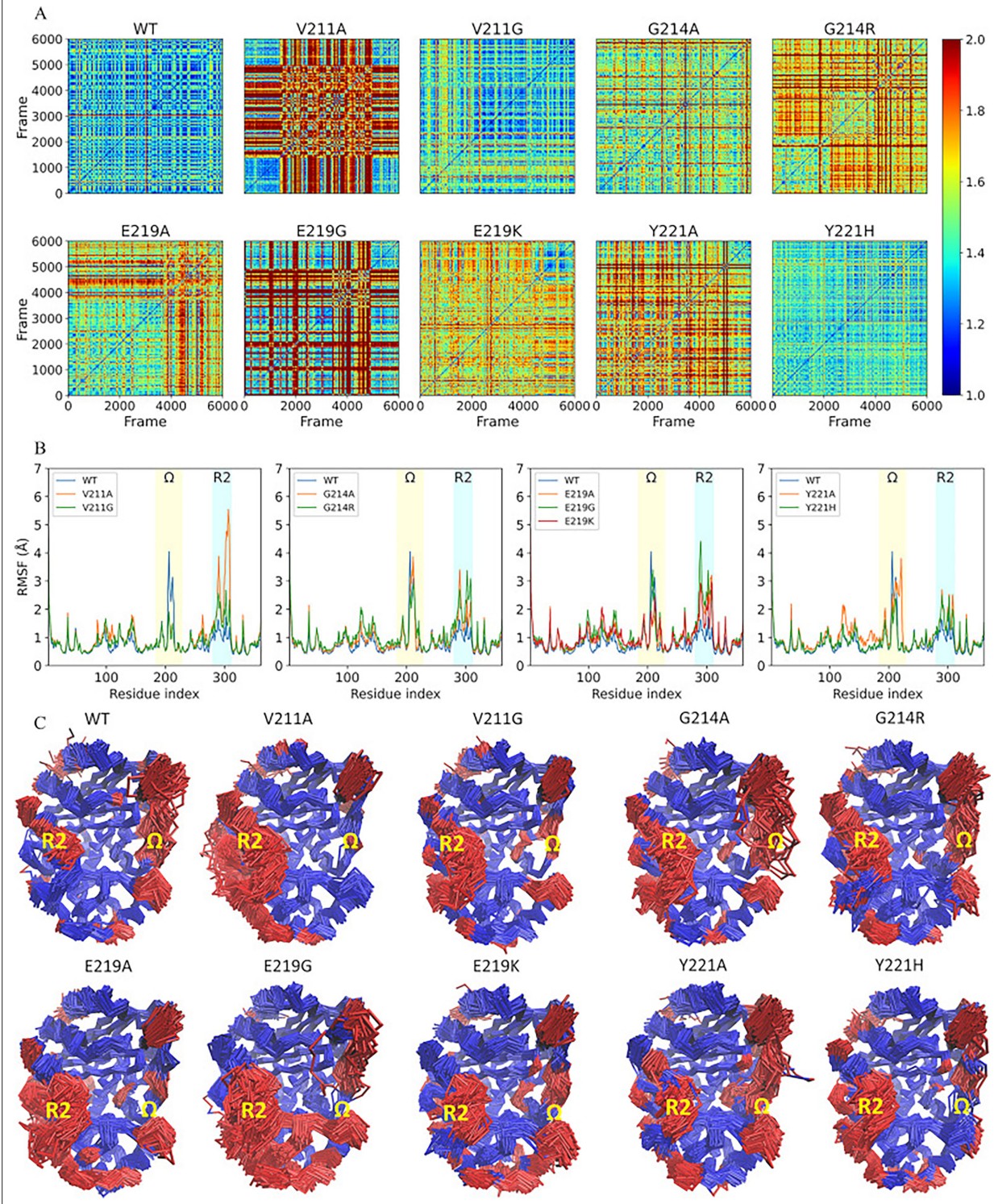

**Figure 2.** Structural stability and dynamic flexibility analyses of wild-type PDC-3 $\beta$-lactamase and its variants. (**A**) Pairwise root mean square deviation (RMSD) comparison of wild-type PDC-3 and its variants. The cross-correlation matrix shows the RMSD values between each pair of structures. The color intensity represents the RMSD value, with lower values indicating a higher degree of structural similarity between the structures. (**B**) The root-mean-square fluctuation (RMSF) of wild-type PDC-3 and its variants. The $\Omega$-loop (residues G183 to S226) is highlighted in yellow, and the R2-loop (residues L280 to Q310) is highlighted in blue. (**C**) Core $C_\alpha$ RMSD superimposition of wild-type PDC-3 and its mutants. The blue parts represent the least mobile C$\alpha$ atoms (80%) while the red parts highlight the most mobile atoms (20%).

The online version of this article includes the following figure supplement(s) for figure 2:

*Figure 2 continued on next page*

*Figure 2 continued*

**Figure supplement 1.** The distribution of RMSD values of wild-type PDC-3 and its variants.

**Figure supplement 2.** RMSD as a function of the fraction of the atoms considered in the alignment.

active site cavity to adopt different sizes and shapes, thus affecting the binding of different $\beta$-lactams and allowing for extended-spectrum activity of some class C $\beta$-lactamases.

## E219K and Y221A mutations facilitate proton transfer

The utilization of Markov state models (MSMs) enabled the analysis of long-term conformational alterations of wild-type PDC-3 and its variants by filtering out local fluctuations related to thermal motion and focusing on underlying conformational transformations (*Bowman et al., 2009*; *Husic and Pande, 2018*; *Scherer et al., 2015*; *Trendelkamp-Schroer and Noé, 2013*). Previous analyses demonstrated that, in addition to the catalytic site, the most significant structural changes occur in the $\Omega$- and R2-loops. Consequently, hydrogen bonds and salt bridges in those loops and in the catalytic site were identified for MSM construction. Distances for all relevant interactions were computed in (i) the active motifs ($S^{64}XXK^{67}$, $Y^{150}SN^{152}$, $K^{315}TG^{317}$), (ii) the $\Omega$-loop (residues G183–S226), and (iii) the R2-loop (residues L280–Q310). To establish the correlation between structural dynamics and active-site pocket, correlation coefficients were computed between the distances of these interactions and the volume of the active-site pockets. A correlation coefficient exceeding 0.3 or falling below –0.3 indicates a positive or negative relationship, respectively. Only the distances of salt bridges and hydrogen bonds that exhibited a positive or negative relationship with the volume of the active-site pockets were selected as features to construct the MSMs. This resulted in the selection of 8 salt bridges and 24 hydrogen bonds (*Figure 3*, *Figure 3—figure supplement 1*).

Inspection of the MSM stationary distributions reveals that E219K and Y221A prominently occupy metastable conformations in which the K67-centered tridentate hydrogen-bond network is fully disrupted, with K67(NZ)–S64(OG), K67(NZ)–N152(OD1), and K67(NZ)–G220(O) all broken (*Figures 3 and 4A*, *Figure 4—figure supplements 1–3*). Notably, this 'fully broken' configuration appears only in E219K and Y221A variants (states 1, 6, and 7 in E219K, and state 3 in Y221A). These three hydrogen bonds could potentially have implications for the catalytic activity of the enzyme, as S64 is a catalytic residue involved in the acylation step of the $\beta$-lactamase. K67 is believed to act as a general base in the acylation step of the $\beta$-lactamase catalytic mechanism, abstracting a proton from the hydroxyl group of S64, which in turn facilitates the nucleophilic attack of the $\beta$-lactam ring (*Tripathi and Nair, 2013*; *Tripathi and Nair, 2016*). Therefore, K67 is thought to toggle between protonated and deprotonated states to facilitate proton transfer in the catalytic cycle (*Figure 1B*). However, when K67 is involved in persistent and energetically favored hydrogen bonding interactions with S64, N152, and G220, these stable interactions can lock it in the protonated state. By contrast, the fully disrupted tridentate network adopted in E219K and Y221A should alleviate this conformational constraint, enabling K67 to more readily undergo the protonation state toggling required for catalysis.

The protonation state ($pK_a$) of K67 in the enzyme is therefore critical: a lowered $pK_a$ could allow K67 to exist as a neutral general base at physiological pH, ready to accept a proton in catalysis (*Chen et al., 2009*). Indeed, analogies from related enzymes suggest that catalytic lysines often have depressed $pK_a$ values (e.g., K47 in PBP5 and K73 in TEM-1) to enable catalytic function (*Golemi-Kotra et al., 2004*; *Zhang et al., 2007*; *Shi et al., 2008*; *Meroueh et al., 2005*). However, directly measuring or computing the $pK_a$ of a buried lysine in a large enzyme is challenging. Constant pH molecular dynamics simulations (CpHMD) provide a powerful *in silico* approach to estimate $pK_a$ by allowing protonation states to fluctuate according to a chosen pH (*Kim et al., 2015*). Here, we employed CpHMD to compute the $pK_a$ of K67 in wild-type PDC-3 and compare it with the E219K and Y221A variants. Titration curves generated from pH-replicated simulations were analyzed to extract K67 $pK_a$ values. Our results indicate that, in wild-type PDC-3, K67 exhibits a $pK_a$ in the range of approximately 8.50–8.79 (*Figure 4*). By contrast, the E219K mutation dramatically reduces the $pK_a$ of K67 to approximately 6.32–6.71, causing K67 to be predominantly deprotonated (neutral) at physiological pH. This deprotonated form is conducive to K67 functioning as a general base readily accepting a proton and thereby facilitating the nucleophilic attack of the S64 hydroxyl group on the $\beta$-lactam ring (*Tripathi and Nair, 2013*; *Tripathi and Nair, 2016*). The Y221A mutation also shifts $pK_a$ of K67

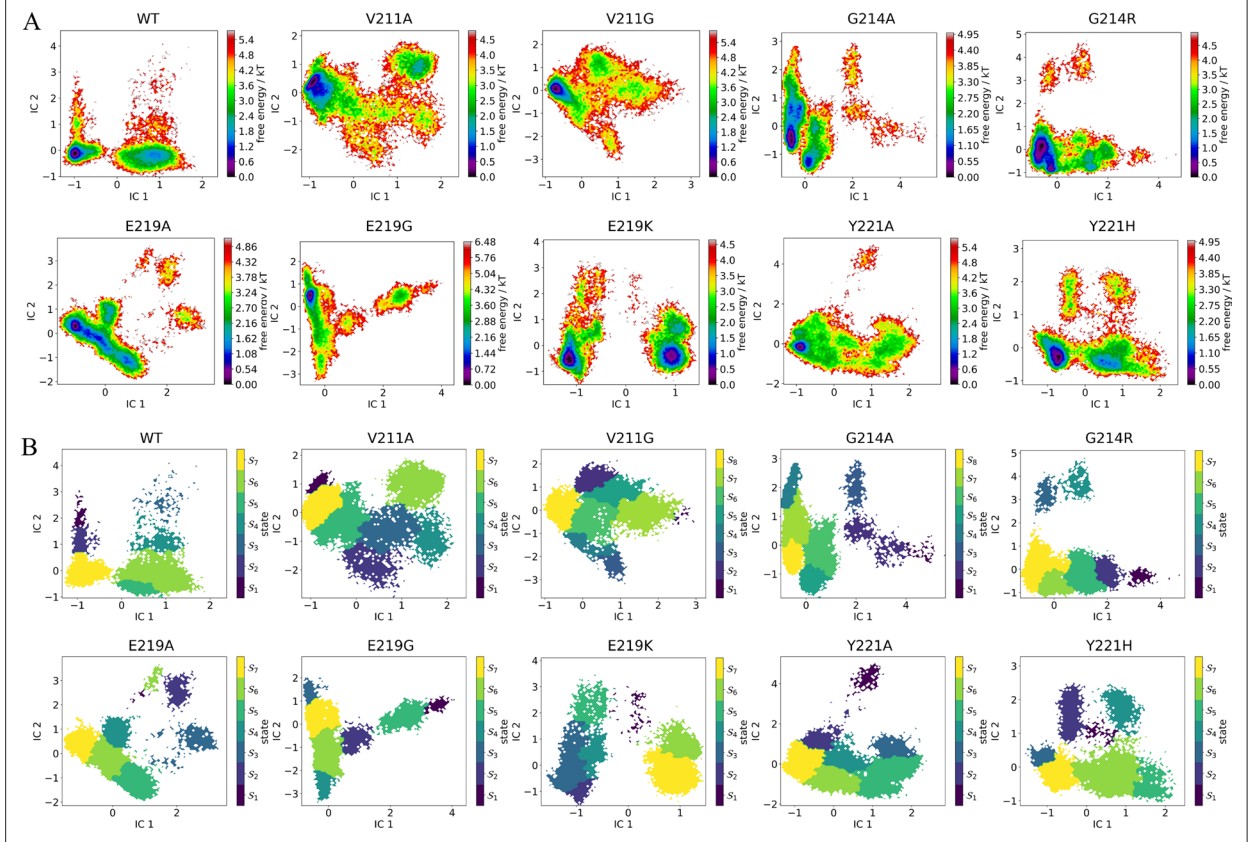

**Figure 3.** Free energy landscapes and metastable-state distributions from Markov state models reveal key conformational transitions in wild-type PDC-3 and its variants. (**A**) The free energy landscape for the microstates of the wild-type PDC-3 and its mutants. (**B**) The metastable states grouped from microstates of PDC-3 and its variants systems. The microstates were grouped by the PCCA method into metastable states in all systems.

The online version of this article includes the following source data and figure supplement(s) for figure 3:

**Source data 1.** The mean first passage time (MFPT) estimates.

**Figure supplement 1.** Correlation coefficients between the distances of key interactions (32 features) and volume of active-site pockets.

**Figure supplement 2.** The convergence behavior of the implied timescales related to the first 10 slowest processes.

**Figure supplement 3.** The Chapman–Kolmogorov test plot of wild-type PDC-3.

**Figure supplement 4.** The Chapman–Kolmogorov test plot of V211A variant.

**Figure supplement 5.** The Chapman–Kolmogorov test plot of V211G variant.

**Figure supplement 6.** The Chapman–Kolmogorov test plot of G214A variant.

**Figure supplement 7.** The Chapman–Kolmogorov test plot of G214R variant.

**Figure supplement 8.** The Chapman–Kolmogorov test plot of E219A variant.

**Figure supplement 9.** The Chapman–Kolmogorov test plot of E219G variant.

**Figure supplement 10.** The Chapman–Kolmogorov test plot of E219K variant.

**Figure supplement 11.** The Chapman–Kolmogorov test plot of Y221A variant.

**Figure supplement 12.** The Chapman–Kolmogorov test plot of Y221H variant.

down (7.60–8.06), though to a lesser degree than E219K. As previously noted, the E219K and Y221A mutations weaken the tridentate hydrogen-bond networks (K67(NZ)-S64(OG), K67(NZ)-N152(OD1), K67(NZ)-G220(O)), thereby enabling K67 to more flexibly adjust both its conformation and its protonation state, which in turn promotes more efficient proton transfer. Experimentally, these mutations also confer increased sensitivity to cephalosporin antibiotics, which aligns with the conformational and protonation-state shifts observed in the simulations (*Barnes et al., 2018*). Collectively, these findings

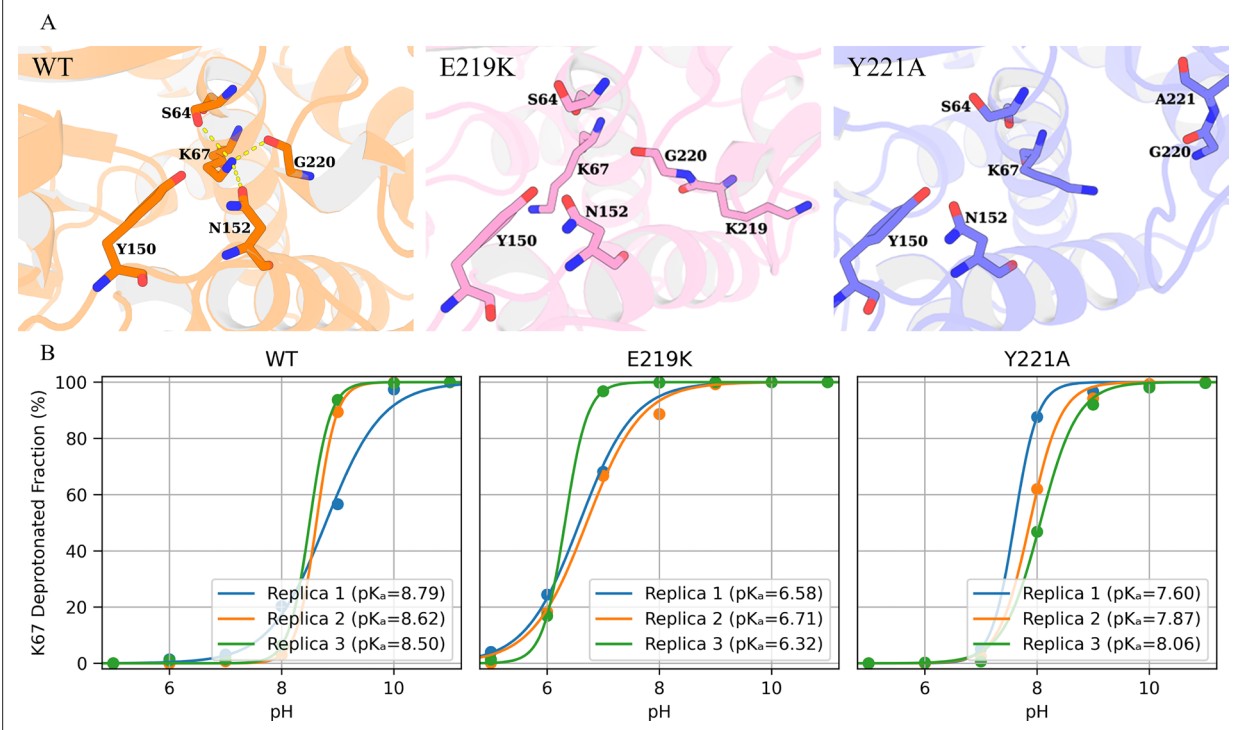

**Figure 4.** E219K and Y221A mutations reshape the catalytic conformations and protonation states of K67. (**A**) Hydrogen bond interactions (dashed lines) between K67(NZ)-S64(OG), K67(NZ)-N152(OD1), and K67(NZ)-G220(O) are formed in wild-type PDC-3 (orange) but broken in the E219K (pink) and Y221A (blue) variants. (**B**) The pH titration curves for K67 based on three replicate constant pH MD simulations. Each point indicates the fraction of deprotonated K67 at a given pH, and the lines are best fits to a titration model. The estimated $pK_a$ values are shown in the legend.

The online version of this article includes the following figure supplement(s) for figure 4:

**Figure supplement 1.** TICA plot illustrates the distribution of wild-type PDC-3 and its variants with the color indicating the K67(NZ)-S64(OG) distance.

**Figure supplement 2.** TICA plot illustrates the distribution of wild-type PDC-3 and its variants with the color indicating the K67(NZ)-N152(OD1) distance.

**Figure supplement 3.** TICA plot illustrates the distribution of wild-type PDC-3 and its variants with the color indicating the K67(NZ)-G220(O) distance.

**Figure supplement 4.** Time-resolved deprotonation of K67 in WT, E219K, and Y221A over 200-ns constant-pH MD simulations at six pH values (5, 6, 7, 8, 9, 10, and 11).

reveal that the E219K and Y221A substitutions disrupt the tridentate hydrogen-bond network, which lowers the $pK_a$ of K67 and enhances its ability to act as a general base. This elevated proton-transfer efficiency, in turn, improves the enzyme's catalytic performance.

Moreover, the mean first passage time (MFPT) data indicate that once the E219K variant forms one of these bond-broken states, it remains there for thousands of nanoseconds (8,262.0±2,573.0 ns to 12,769.0±3327.0 ns) before transitioning to a bond-formed state (state 3) (**Figure 3** and **Figure 3—source data 1**). Likewise, the reverse process also occurs on a microsecond timescale, demonstrating that both directions are kinetically stabilized in E219K. As a result, E219K displays two dominant energy minima in its free-energy landscape, whereas other variants typically show only one (**Figure 3A**). Prolonged residence in a bond-broken conformation implies that K67 is more likely to remain deprotonated, enhancing catalytic function. By contrast, in Y221A, the equivalent bond-broken state (state 3) shifts more readily into other metastable states, including the most stable state 7 (bond-formed), in only 780.2±46.8 ns. Although the reverse transition from bond-formed to bond-broken in Y221A requires a somewhat longer 1,737.6±139.7 ns, this timescale remains far shorter than E219K's multi-microsecond range. Consequently, Y221A can dynamically switch between 'formed' and 'broken' conformations with much greater ease. This difference in conformational kinetics helps explain differences in how each mutant enhances hydrolysis rates. E219K achieves it through stable, long-lived 'active' conformations, while Y221A relies on faster switching and conformational plasticity.

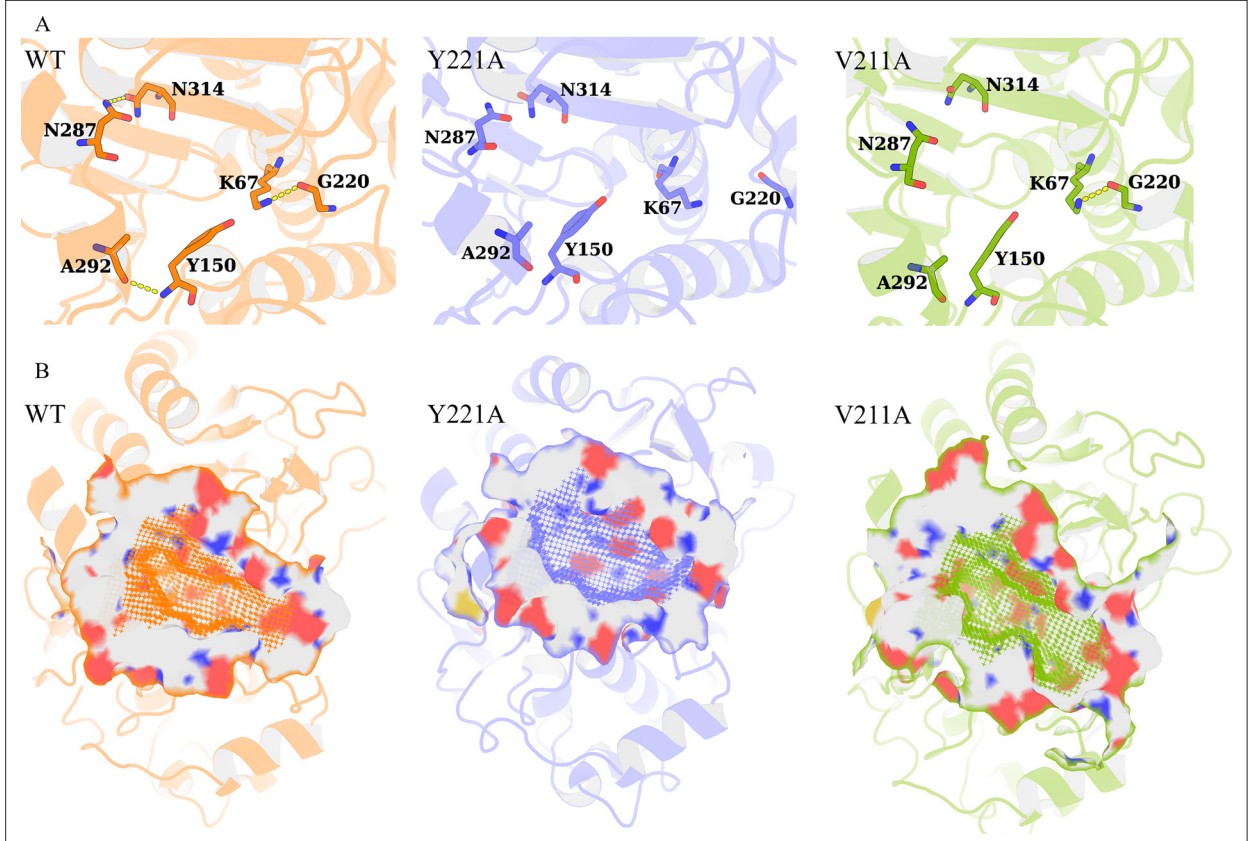

**Figure 5.** Structural visualization of the enlarged active-site pocket. (**A**) The K67(NZ)-G220(O), Y150(N)-A292(O), and N287(ND2)-N314(OD1) interactions in representative structures of wild-type PDC-3, Y221A, and V211A variants. The yellow dashed lines represent the interactions. (**B**) The active-site pockets of wild-type PDC-3, Y221A, and V211A variants are shown as surface representation.

The online version of this article includes the following figure supplement(s) for figure 5:

**Figure supplement 1.** TICA plot illustrates the distribution of wild-type PDC-3 and its variants with the color indicating the Y150(N)-A292(O) distance.

**Figure supplement 2.** TICA plot illustrates the distribution of wild-type PDC-3 and its variants with the color indicating the N287(ND2)-N314(OD1) distance.

## Substitutions enlarge the active-site pocket to accommodate bulkier R1 and R2 groups of $\beta$-lactams

In addition to facilitating catalytic proton transfer, $\Omega$-loop substitutions also remodel the steric architecture of the active site. Specifically, the K67–G220 hydrogen bond discussed above may directly influence the shape and volume of the R1 side of the binding cavity. K67 is part of the conserved catalytic motif S$^{64}$XXK$^{67}$, whereas G220 resides within the $\Omega$-loop (*Figure 1D*). Likewise, A292 and N287 are located on the R2-loop, while Y150 and N314 are also located in the catalytic motifs. The Y150(N)–A292(O) and N287(ND2)–N314(OD1) interactions are therefore proposed to regulate the space available on the R2 side of the pocket. Importantly, the MSM-derived metastable states separate into basins in which these contacts remain formed and basins in which they are broken (*Figure 4—figure supplement 3*, *Figure 5—figure supplements 1 and 2*). Because transitions between metastable states occur on slow timescales, this contact switching likely reflects slow loop rearrangements that control the active-site cavity (*Figure 5*).

To validate this hypothesis, the mean pocket volume and the donor–acceptor distances for the three putative hydrogen-bond pairs were computed for each metastable state (*Figure 6*). In wild-type PDC-3, the pocket remains compact across the metastable ensemble. The global free-energy minimum basin of the wild-type landscape (state 7) exhibits a mean volume of 1048.9 ± 143.7 Å$^3$ (*Figures 3A and 6A*). In this state, the three interactions are predominantly consistent with hydrogen-bonding geometry, as reflected by their mean donor–acceptor distances for K67(NZ)–G220(O) (3.2 Å),

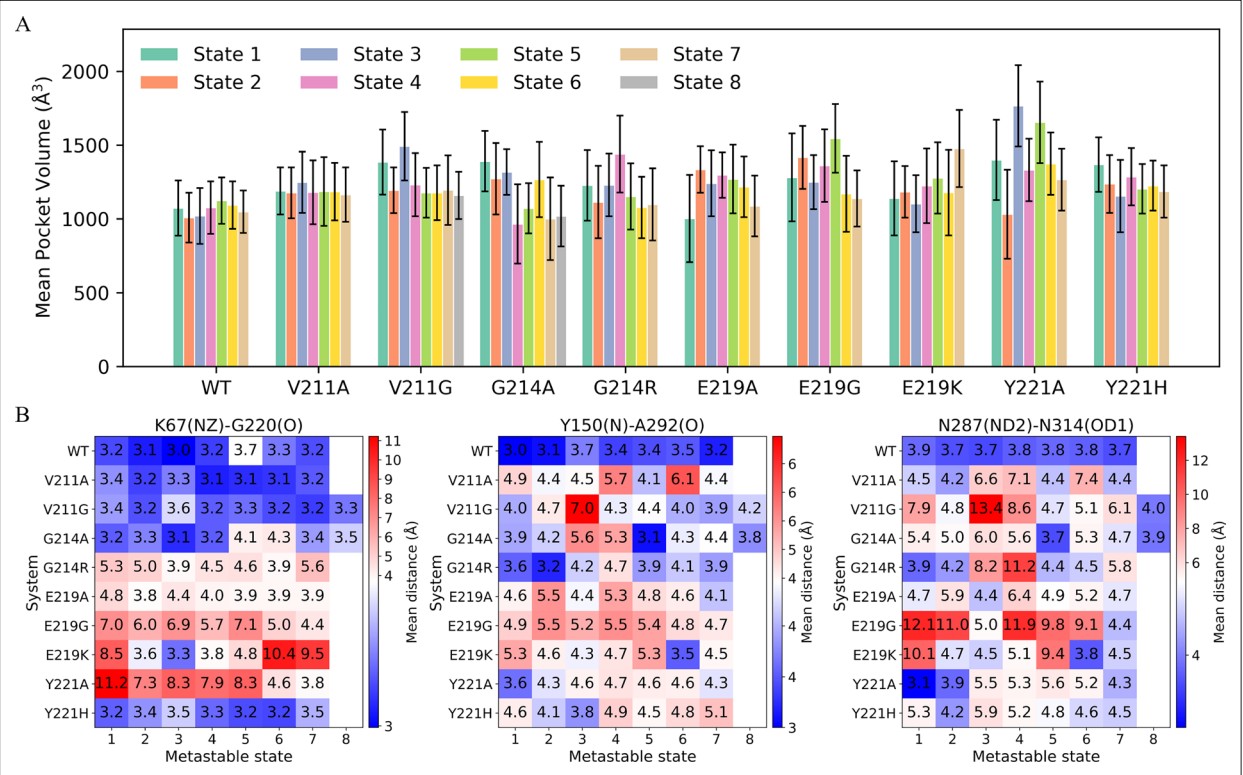

**Figure 6.** Metastable-state pocket volumes and key interaction patterns across wild-type PDC-3 and its variants. (**A**) Mean active-site pocket volume for each metastable state in wild-type PDC-3 and its variants. Bars denote the mean pocket volume (Å³), and error bars indicate the standard deviation across frames assigned to each state. (**B**) Heat maps show the mean distances (Å) for three contacts that may contribute to pocket-volume changes across MSM metastable states (K67(NZ)–G220(O), Y150(N)–A292(O), and N287(ND2)–N314(OD1)). Color encodes the distance magnitude, with blue indicating lower values and red indicating higher values.

The online version of this article includes the following figure supplement(s) for figure 6:

**Figure supplement 1.** Mean distances of three contacts (K67(NZ)–G220(O), Y150(N)–A292(O), and N287(ND2)–N314(OD1)) across metastable states in wild-type PDC-3 and its variants.

Y150(N)–A292(O) (3.2 Å), and N287(ND2)–N314(OD1) (3.7 Å) (*Figure 6B*, *Figure 6—figure supplement 1*). In contrast, Y221A shows pronounced pocket expansion in multiple states. State 3 reaches 1766.8 ± 274.9 Å³ (+68.4% relative to the wild-type global-minimum state, state 7), and state 5 reaches 1654.9 ± 275.8 Å³ (+57.8% relative to the same reference). In both states, the three hydrogen-bond pairs are largely disrupted, with the corresponding donor–acceptor distances substantially increased. This mechanistic mapping provides a direct structural rationale for how single $\Omega$-loop substitutions can expand the active-site cavity to accommodate bulkier R1 and R2 groups of $\beta$-lactams (*Figure 5*). Y221A state 1 is also associated with a large pocket volume of 1399.9 ± 272.4 Å³. Although the R2-side pairs N287(ND2)–N314(OD1) and Y150(N)–A292(O) remain consistent with hydrogen-bonding geometry (3.1 Å and 3.6 Å, respectively), the K67(NZ)–G220(O) distance increases to 11.2 Å. Pronounced pocket expansion is also observed in E219G. In E219G state 5, the pocket reaches 1546.1 ± 233.1 Å³ (+47.4% relative to the wild-type global-minimum state, state 7), with all three hydrogen-bond pairs largely disrupted. Although these expansive states are not the global energy minimum in Y221A or E219G (*Figure 3A*), they align with well-defined low free-energy basins, indicating that substantial cavity expansion can be thermodynamically accessible.

A more striking thermodynamic shift is observed for E219K. In this variant, the most expanded-pocket ensemble coincides with a dominant free-energy minimum. State 7 exhibits a large pocket volume of 1477.4 ± 261.3 Å³, corresponding to a 40.9% increase relative to the wild-type global-minimum state. Structurally, this minimum is characterized by a pronounced expansion on the R1 side, with the mean K67(NZ)–G220(O) distance extended to 9.5 Å. In addition, the R2-side hydrogen bonds Y150(N)–A292(O) and N287(ND2)–N314(OD1) are disrupted, with mean donor–acceptor distances

of 4.5 Å and 4.5 Å, respectively. This thermodynamic stabilization of an expanded-pocket minimum provides a plausible mechanistic basis for the pronounced resistance phenotype, in which the E219K mutant shows markedly reduced susceptibility to cephalosporins (*Barnes et al., 2018*).

G214R state 4 and V211G state 3 also sample enlarged active-site cavities. In these states, the expansions arise primarily from outward displacements of the R2-loop, reflected by markedly increased mean N287(ND2)–N314(OD1) and Y150(N)–A292(O) distances, while the R1-side architecture remains comparatively compact. Notably, these expansive conformations occupy sparsely populated, higher-free-energy basins, indicating that they are not strongly stabilized in the apo ensemble. Instead, they likely represent excited-state expansions that are only transiently accessed but can be selectively stabilized upon substrate binding. Thus, ligands with bulky substituents might capture these pre-existing R2-expanded conformations, shifting the ensemble toward a larger cavity and thereby enabling accommodation of larger R2 groups.

## Conclusions

This investigation of the effects of substitutions in the PDC-3 $\beta$-lactamase has provided valuable information on the protein's dynamics. The study indicates that substitutions can have a significant impact on the stability and flexibility of the $\Omega$-loop and R2-loop, both of which are critical for the $\beta$-lactamase function. Specifically, the G214A, G214R, E219G, and Y221A variants, as well as the wild-type PDC-3, exhibit high flexibility in the $\Omega$-loop, while the V211A and E219G variants show the highest flexibility in the R2-loop. Moreover, the hydrogen-bond network around K67—specifically K67(NZ)–S64(OG), K67(NZ)–N152(OD1), and K67(NZ)–G220(O)—governs the proton-transfer step essential for $\beta$-lactam hydrolysis. When these three bonds remain intact, K67 tends to stay protonated, limiting its availability to accept a proton from S64. Mutations such as E219K and Y221A disrupt this tridentate network, reducing K67's $pK_a$ and facilitating efficient proton transfer in hydrolysis. Additionally, K67(NZ)–G220(O), Y150(N)–A292(O), and N287(ND2)–N314(OD1) interactions further modulate R1/R2-loop conformations. Breaking these hydrogen bonds typically shifts the active site to a more expansive configuration, accommodating larger cephalosporin substrates. Overall, the findings of this study provide significant insight into the dynamics of the PDC-3 $\beta$-lactamase, revealing the critical roles played by the $\Omega$-loop and R2-loop in its function. These insights gained from this study will aid in the design of more potent antibiotics and $\beta$ inhibitors for treating bacterial infections.

## Methods
### Initial structure preparation

All-atom MD simulations of wild-type PDC-3 and its variants were performed. First, the simulations of their conformations were initiated from the X-ray crystallographic structure (PDB ID: 4HEF) at 1.86 Å, after modification of T79A (*Lahiri et al., 2013*). PDC-3 wild-type and nine variants (V211A, V211G, G214A, G214R, E219A, E219G, E219K, Y221A, and Y221H) were constructed *in silico* using the ICM mutagenesis program (*Abagyan et al., 1994*). To ensure the accurate protonation states of the protein, PROPKA 3.0 was employed to assign the protonation states of N-terminus, C-terminus, cationic residues, and anionic residues based on a neutral pH local environment (*Olsson et al., 2011*). In addition, all acidic residues were negatively charged, while alkaline Lys and Arg residues remained positively charged. His was protonated based upon the suggestion by PROPKA 3.0 analysis and also checked by visual inspection.

### AdaptiveBandit simulations

AdaptiveBandit MD (AB-MD) simulation is a reinforcement learning-based enhanced sampling method that offers a more efficient exploration of the protein's conformational space while maintaining unbiased, thermodynamically accurate ensembles (*Pérez et al., 2020*). The advantage of using AB-MD is that it does not alter the underlying potential energy surface, it retains physically realistic dynamics and eliminates the need for reweighting of biased trajectories. As a result, AB-MD can attain a similar or greater depth of conformational sampling with significantly less total simulation time (i.e. lower computational cost) than either extended conventional MD or other enhanced sampling approaches.

The WT and variant structures served as the starting point for subsequent molecular dynamics (MD) simulation. Multi-microsecond MD simulations of wild-type PDC-3 and its variants were conducted

using the Amberff14SB force field (*Maier et al., 2015*). All simulations were run using the ACEMD engine (*Doerr et al., 2016*; *Harvey et al., 2009*). Each structure was solvated in a pre-equilibrated periodic cubic box of water molecules represented by the three-point charge TIP3P model, whose boundary is at least 10 Å from any atoms so that the protein does not interact with its periodic images. Periodic boundary conditions in all directions were utilized to reduce finite system size effects. The potassium ions were added to make each system electrically neutral. Long-range electrostatic interactions were computed using the particle mesh Ewald summation method (*Cerutti et al., 2009*). Subsequently, each system was energy minimized for 5000 steps by conjugate gradient to remove any local atomic clashes and then equilibrated for 5 ns at 1 atmospheric pressure using Berendsen barostat (*Feenstra et al., 1999*).

Initial velocities within each simulation were sampled from the Maxwell–Boltzmann distribution at a temperature of 300 K. Simulations were performed in the NVT ensemble using a Langevin thermostat with a damping of 0.1 ps-1 and hydrogen mass repartitioning scheme to achieve time steps of 4 fs. Multiple short MSM-based adaptively sampled simulations were run using the ACEMD molecular dynamics engine (*Doerr et al., 2016*; *Harvey et al., 2009*). The standard adaptive sampling algorithm performs several rounds of short parallel simulations. To avoid any redundant sampling, the algorithm generates a Markov state model (MSM) and uses the stationary distribution of each state to obtain an estimate of its free energy. It then selects any sampled conformation from a low free energy stable state and respawns a new round of simulations. In this context, the MetricSelfDistance function was set to consider the number of native Cα contacts formed for all residues, which were then used to build the MSMs. The exploration value was 0.01 and goal-scoring function was set to 0.3. For each round, 4 simulations of 300 ns were run in parallel until the cumulative time exceeded 30 μs. The trajectory frames were saved every 0.1 ns. 100 trajectories for each system were collected with each trajectory counting 3000 frames.

## Constant pH molecular dynamics

To investigate the protonation behavior of K67 in wild-type PDC-3 and its E219K and Y221A variants, K67, Y150, E/K219, and K315 were selected as titratable sites in CpHMD simulations because they lie in proximity to the site of interest (K67) and together form its immediate electrostatic network (*Kim et al., 2015*; *Lahiri et al., 2013*). The starting structures were identical to those used for AB-MD. The simulations were performed in the Amber suite with the ff99SB force field and an implicit Generalized Born (GB) solvent model (*Maier et al., 2015*; *Mongan et al., 2004*). The protein–solvent complex was energy-minimized for a total of 5000 steps—10 steps of steepest-descent followed by 4990 steps of conjugate-gradient—with harmonic positional restraints (10 kcal/mol·Å²) on the backbone atoms to relax side-chain clashes (*Brooks et al., 1983*; *Schlegel, 1982*). The system was then heated from 10 K to 300 K over 1 ns, followed by another 1 ns of equilibration at 300 K under Langevin dynamics Schneider and Stoll (1978), with a 2 fs time step and SHAKE constraints on hydrogen-containing bonds. During heating, a weaker restraint (2 kcal/mol·Å²) was applied to the backbone atoms to maintain the overall fold while allowing side-chain relaxation, and protonation states were kept fixed in this phase. Equilibrium simulations under constant pH conditions were subsequently conducted at pH 5, 6, 7, 8, 9, 10, and 11 by periodically (every 10 steps) attempting protonation-state changes via a Monte Carlo protocol (*Kim et al., 2015*). Each pH condition consisted of 50 ns of equilibration followed by 200 ns of production. To confirm convergence, the deprotonation fraction of each residue was monitored over time and found to reach a stable plateau within the final portion of the production simulations. Consequently, the last 50 ns of production for each pH value were used to calculate the deprotonation fractions, ensuring that the analyzed region reflected a converged state. Each system was simulated in triplicate, ultimately providing consistent K67 $pK_a$ estimates for the wild-type, E219K, and Y221A variants. All analyses were done with AmberTools *cphstats* and in-house Python scripts (*Case et al., 2023*).

## Markov state models

The PyEMMA software (version 2.5.9) was employed to construct the Markov state models (*Husic and Pande, 2018*; *Scherer et al., 2015*). The software determines the kinetically relevant metastable states and their interconversion rate from all trajectories of the all-atom molecular dynamics of the wild-type PDC-3 and its variants. Firstly, to evaluate the MSM construction, the conformations defining each

frame of the MD trajectories were converted into an intuitive basis. In this step, the features that can represent the slow dynamical modes of these systems were selected. Then, the conformational space was projected to a two-dimensional space using time-lagged independent component analysis (TICA) (*Pérez-Hernández and Noé, 2016*). Using the *k*-means clustering technique, all conformations from MD simulations were grouped into microstates based on the TICA embedding (*Peng et al., 2018*). The conformations in the same cluster are geometrically similar and interconvert quickly. After that, the transition matrix between the microstates was built using Bayesian estimation at the appropriate lag time (*Trendelkamp-Schroer and Noé, 2013*). The lag time was selected where the implied time scales converged, and the transitions between the microstates became the Markovian process. Each indicated time scale represents the average transition time between two groups of states. The microstates were then clustered into a few metastable states using Perron cluster cluster analysis (PCCA) based on their kinetic similarities (*Bowman et al., 2009*). Additionally, the Chapman–Kolmogorov (CK) test was performed to validate the constructed model further (*Barendregt et al., 2019*). The CK test measures the reliability of the Markov state models by comparing the predicted residence probability of each microstate obtained from MSMs with those directly computed from MD simulations at longer timescales. Furthermore, the free energies for each metastable state ($S_i$) were computed from its stationary MSM probability $\pi$ using the relation:

$$\Delta G(S_i) = -k_B T \ln(\sum_{j \in S_i} \pi_j)$$

(1)

where $\pi_j$ denotes the MSM stationary weight of the $j$th microstate, $k_B$ is the Boltzmann constant, and $T$ is the temperature. Subsequently, the MFPT out of and into the macrostate $S_i$ were computed using the Bayesian MSM (*Polizzi et al., 2016*).

## Structural analysis

MDTraj is a robust software package that facilitates the analysis of molecular dynamics (MD) simulations by enabling the manipulation of MD trajectory data from a variety of files (*McGibbon et al., 2015*). The package provides Python-based tools that allow for the efficient computation of structural and dynamic properties of biomolecules. In this study, MDTraj was utilized to compute several important metrics that are critical in the analysis of MD simulations. Specifically, we used MDTraj to calculate the root-mean-square-fluctuations (RMSF) and root-mean-square-deviation (RMSD) of the protein structure, hydrogen bonds, and salt bridges. RMSF quantifies the average positional fluctuation of the Cα atom of each residue during the MD simulation relative to its position in the equilibrated reference structures. RMSD quantifies the average displacement of the protein's Cα atoms from their positions in an equilibrated reference structure. In addition, MDLovoFit was used to show the graphical representation of RMSD results (*Martínez, 2015*). Furthermore, pairwise RMSD analyses were performed using the pytraj package, which allowed us to assess the structural similarities and differences among the conformations sampled during the simulation (*Roe and Cheatham, 2013*). Hydrogen bonds were defined as a distance of less than 3.5 Å between a hydrogen bond donor and acceptor, with a hydrogen-donor-acceptor angle greater than 30°. Salt bridges were defined as a distance of less than 4.0 Å between a positively charged amino acid side chain (lysine or arginine) and a negatively charged side chain (aspartate or glutamate). The volume of the active-site pocket was calculated using ParkVFinder (*Guerra et al., 2020*). Visualization of the structures of protein was performed using PyMOL (*Schrödinger and DeLano, 2020*).

## Acknowledgements

Research reported herein was supported in part by funds the National Institute of Allergy and Infectious Diseases of the National Institutes of Health under Award Number R01AI063517 to RAB and SH. The content is solely the responsibility of the authors and does not necessarily represent the official views of the Department of Veterans Affairs or the National Institutes of Health.

## Additional information

### Competing interests

Robert A Bonomo: RAB reports grants from Merck, Wockhardt, Shionogi, and Venatorx outside the submitted work. Shozeb Haider: Reviewing editor, eLife. The other authors declare that no competing interests exist.

### Funding

| Funder | Grant reference number | Author |
|---|---|---|
| National Institute of Allergy and Infectious Diseases | R01AI063517 | Robert A Bonomo |

The funders had no role in study design, data collection and interpretation, or the decision to submit the work for publication.

### Author contributions

Shuang Chen, Formal analysis, Investigation, Visualization, Methodology, Writing - original draft; Andrew R Mack, Formal analysis, Investigation, Writing - original draft; Andrea M Hujer, Validation, Investigation; Christopher R Bethel, Validation, Investigation, Writing - original draft; Robert A Bonomo, Supervision, Funding acquisition, Validation, Writing – review and editing; Shozeb Haider, Conceptualization, Supervision, Project administration, Writing – review and editing

### Author ORCIDs

Shuang Chen ⓘ https://orcid.org/0009-0005-4968-5869
Shozeb Haider ⓘ https://orcid.org/0000-0003-2650-2925

Reviewer #2 (Public review): https://doi.org/10.7554/eLife.107688.3.sa1
Reviewer #3 (Public review): https://doi.org/10.7554/eLife.107688.3.sa2
Author response https://doi.org/10.7554/eLife.107688.3.sa3

## Additional files

### Supplementary files

MDAR checklist

### Data availability

All files required to run the simulations (topology, coordinates, input), processed trajectories (xtc), corresponding coordinates (pdb), metastable PDB files for each system described in this manuscript can be downloaded from the DOI: https://doi.org/10.57760/sciencedb.15876.

The following dataset was generated:

| Author(s) | Year | Dataset title | Dataset URL | Database and Identifier |
|---|---|---|---|---|
| Haider S | 2024 | Ω-Loop mutations control the dynamics of the active site by modulating a network of hydrogen bonds in PDC-3 β-lactamase | https://doi.org/10.57760/sciencedb.15876 | Science DataBank, 10.57760/sciencedb.15876 |

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
