## [Editor Report · eLife Assessment]

This article uses adaptive-bandit simulations to describe the dynamics of the *Pseudomonas*-derived chephalosporinase PDC-3 β-lactamase and its mutants to better understand antibiotic resistance. The finding that clinically observed mutations alter the flexibility of the Ω- and R2-loops, reshaping the cavity of the active site, is **valuable** to the field. The evidence is considered **incomplete**, however, with the need for analysis to demonstrate equilibrium weighting of adaptive trajectories and related measures of statistical significance.

---

## [Referee Report · Reviewer #2 (Public review)]

Summary:

In the manuscript entitled "Ω-Loop mutations control dynamics 2 of the active site by modulating the 3 hydrogen-bonding network in PDC-3 4 β-lactamase", Chen and coworkers provide a computational investigation of the dynamics of the enzyme Pseudomonas-derived chephalosporinase 3 (PDC3) and some mutants associated with increased antibiotic resistance. After an initial analysis of the enzyme dynamics provided by RMSD/RMSF, the author conclude that the mutations alter the local dynamics within the omega loop and the R2 loop. The authors show that the network of hydrogen bonds in disrupted in the mutants. Constant pH calculations showed that the mutations also change the pKa of the catalytic lysine 67 and pocket volume calculations showed that the mutations expand the catalytic pocket. Finally, time-independent componente analysis (tiCA) showed different profiles for the mutant enzyme as compared to the wild type.

Strengths:

The scope of the manuscript is definitely relevant. Antibiotic resistance is an important problem and, in particular, *Pseudomonas aeruginosa* resistance is associated with an increasing number of deaths. The choice of the computational methods is also something to highlight here. Although I am not familiar with Adaptive Bandit Molecular Dynamics (ABMD), the description provided in the manuscript that this simulation strategy is well suited for the problem under evaluation.

Weaknesses:

In the revised version, the authors addressed my concerns regarding their use of the MSM, and in my view, their conclusions are now much more robust and well-supported by the data. While it would be very interesting to see a quantitative correlation between the effects of the mutations observed in the MD data and relevant experimental findings, I understand that this may be beyond the scope of the manuscript.

---

## [Referee Report · Reviewer #3 (Public review)]

Summary:

This manuscript aims to explore how mutations in the PDC-3 3 β-lactamase alter its ability to bind and catalyse reactions of antibiotic compounds. The topic is interesting and the study uses MD simulations and to provide hypotheses about how the size of the binding site is altered by mutations that change the conformation and flexibility of two loops that line the binding pocket. Some greater consideration of the uncertainties and how the method choice affect the ability to compare equilibrium properties would strengthen the quantitative conclusions. While many results appear significant by eye, quantifying this and ensuring convergence would strengthen the conclusions.

Strengths:

The significance of the problem is clearly described the relationship to prior literature is discussed extensively.

Comments on revised version:

I am concerned that the authors state in the response to reviews that it is not possible to get error bars on values due to the use of the AB-MD protocol that guides the simulations to unexplored basins. Yet the authors want to compare these values between the WT and mutants. This relates to RMSD, RMSF, % H-bond and volume calculations. I don't accept that you cannot calculate an uncertainty on a time averaged property calculated across the entire simulation. In these cases you can either run repeat simulations to get multiple values on which to do statistical analysis, or you can break the simulation into blocks and check both convergence and calculate uncertainties.

I note that the authors do provide error bars on the volumes, but the statistics given for these need closer scrutiny (I cant test this without the raw data). For example the authors have p<0.0001 for the following pair of volumes 1072 {plus minus} 158 and 1115 {plus minus} 242, or for SASA p<0.0001 is given for 2 identical numbers 155+/- 3.

I also remain concerned about comparisons between simulations run with the AB-MD scheme. While each simulation is an equilibrium simulation run without biasing forces, new simulations are seeded to expand the conformational sampling of the system. This means that by definition the ensemble of simulations does not represent and equilibrium ensemble. For example, the frequency at which conformations are sampled would not be the same as in a single much longer equilibrium simulation. While you may be able to see trends in the differences between conditions run in this way, I still don't understand how you can compare quantitative information without some method of reweighing the ensemble. It is not clear that such a rewieghting exists for this methods, in which case I advise some more caution in the wording of the comparisons made from this data.

At this stage I don't feel the revision has directly addressed the main comments I raised in the earlier review, although there is a stronger response to the comments of Reviewer #2.

---

## [Author Response]

The following is the authors’ response to the current reviews.

**Reviewer #2 (Public review):**
Summary:In the manuscript entitled "Ω-Loop mutations control dynamics 2 of the active site by modulating the 3 hydrogen-bonding network in PDC-3 4 β-lactamase", Chen and coworkers provide a computational investigation of the dynamics of the enzyme Pseudomonas-derived chephalosporinase 3 (PDC3) and some mutants associated with increased antibiotic resistance. After an initial analysis of the enzyme dynamics provided by RMSD/RMSF, the author conclude that the mutations alter the local dynamics within the omega loop and the R2 loop. The authors show that the network of hydrogen bonds in disrupted in the mutants. Constant pH calculations showed that the mutations also change the pKa of the catalytic lysine 67 and pocket volume calculations showed that the mutations expand the catalytic pocket. Finally, time-independent componente analysis (tiCA) showed different profiles for the mutant enzyme as compared to the wild type.Strengths:The scope of the manuscript is definitely relevant. Antibiotic resistance is an important problem and, in particular, *Pseudomonas aeruginosa* resistance is associated with an increasing number of deaths. The choice of the computational methods is also something to highlight here. Although I am not familiar with Adaptive Bandit Molecular Dynamics (ABMD), the description provided in the manuscript that this simulation strategy is well suited for the problem under evaluation.Weaknesses:In the revised version, the authors addressed my concerns regarding their use of the MSM, and in my view, their conclusions are now much more robust and well-supported by the data. While it would be very interesting to see a quantitative correlation between the effects of the mutations observed in the MD data and relevant experimental findings, I understand that this may be beyond the scope of the manuscript.

Thank you for the careful evaluation and constructive comments. Regarding the suggestion of a more quantitative correlation with experimental observables, we agree that this would be valuable, and we have noted it as an important direction for future work.

**Reviewer #3 (Public review):**
Summary:This manuscript aims to explore how mutations in the PDC-3 3 β-lactamase alter its ability to bind and catalyse reactions of antibiotic compounds. The topic is interesting and the study uses MD simulations and to provide hypotheses about how the size of the binding site is altered by mutations that change the conformation and flexibility of two loops that line the binding pocket. Some greater consideration of the uncertainties and how the method choice affect the ability to compare equilibrium properties would strengthen the quantitative conclusions. While many results appear significant by eye, quantifying this and ensuring convergence would strengthen the conclusions.Strengths:The significance of the problem is clearly described the relationship to prior literature is discussed extensively.Comments on revised version:I am concerned that the authors state in the response to reviews that it is not possible to get error bars on values due to the use of the AB-MD protocol that guides the simulations to unexplored basins. Yet the authors want to compare these values between the WT and mutants. This relates to RMSD, RMSF, % H-bond and volume calculations. I don't accept that you cannot calculate an uncertainty on a time averaged property calculated across the entire simulation. In these cases you can either run repeat simulations to get multiple values on which to do statistical analysis, or you can break the simulation into blocks and check both convergence and calculate uncertainties.

We thank the reviewer for raising this point. We would like to clarify that we did not intend to state that error bars are impossible to obtain under AB-MD. In fact, we reported error bars for several quantities derived from the AB-MD trajectories (we also broke the trajectories into blocks and calculated uncertainties for RMSF in our first-round response as you suggested). However, these data are closely related to your concern about comparing quantitative information without an appropriate reweighting of the ensemble. Therefore, in the revised manuscript, we removed quantitative analyses that were calculated directly from the raw AB-MD trajectories. Instead, the quantitative comparisons are now obtained from MSM analysis. We report pocket volumes and key interaction metrics for MSM metastable states, with corresponding error bars for these MSM-based quantities (Figure 6 and its supplementary figure).

I note that the authors do provide error bars on the volumes, but the statistics given for these need closer scrutiny (I cant test this without the raw data). For example the authors have p<0.0001 for the following pair of volumes 1072 {plus minus} 158 and 1115 {plus minus} 242, or for SASA p<0.0001 is given for 2 identical numbers 155+/- 3.

Thank you for this comment. As noted above, we have removed the table from the manuscript, and the pocket-volume results together with their error bars are now shown in Figure 6. To address the concern raised here and to avoid making the same mistake in future analyses, we re-examined how the statistics were computed. We believe the very small p-values were caused by treating per-frame MD values as independent observations in two-sample t-tests. Because consecutive MD frames are strongly time-correlated, they do not satisfy the independence assumption, which can greatly overestimate the effective sample size and lead to artificially small p-values. For the SASA, a p < 0.0001 is reported even though both values are shown as 155 ± 3. This is due to rounding, which can hide subtle underlying differences.

I also remain concerned about comparisons between simulations run with the AB-MD scheme. While each simulation is an equilibrium simulation run without biasing forces, new simulations are seeded to expand the conformational sampling of the system. This means that by definition the ensemble of simulations does not represent and equilibrium ensemble. For example, the frequency at which conformations are sampled would not be the same as in a single much longer equilibrium simulation. While you may be able to see trends in the differences between conditions run in this way, I still don't understand how you can compare quantitative information without some method of reweighing the ensemble. It is not clear that such a rewieghting exists for this methods, in which case I advise some more caution in the wording of the comparisons made from this data.At this stage I don't feel the revision has directly addressed the main comments I raised in the earlier review, although there is a stronger response to the comments of Reviewer #2.

We thank the reviewer for reiterating this important point, and we agree with the underlying concern. Although AB-MD generates unbiased trajectories, the ensemble of simulations does not represent an equilibrium ensemble. As a result, statistics computed by simply concatenating all AB-MD trajectories should not be used for quantitative comparisons. In the original version, we acknowledge that we reported several quantitative descriptors directly from concatenated AB-MD frames, including (i) distributions of χ1 torsions, (ii) mean pocket volumes and SASA, and (iii) percentages of some key interactions. We agree that this was not appropriate given the adaptive sampling protocol. In the revised manuscript, we have removed these quantitative analyses.

We retained RMSD and RMSF analyses, but we have revised their wording and clarified their purpose. RMSD and RMSF are used only to summarize the structural variability and residue-level mobility observed across the collected trajectory segments and to motivate the selection of structural features for MSM construction. The manuscript now states: “Because AB-MD adaptively seeds new unbiased trajectories to expand conformational sampling, RMSD and RMSF are used here to summarize the structural variability and per-residue mobility observed across the collected trajectories.”

Regarding the reviewer’s question about reweighting, the Markov state model (MSM) provides a principled framework to obtain the stationary distribution *π* from the transition probability matrix *Tτ*. The resulting *πi* gives the equilibrium weight of each microstate *i*, and the corresponding discrete free energy can be written as *Fi*=−*k*_B_*T*ln(*πi*). PCCA then coarse-grains the microstate space into a small number of metastable states. In the revised manuscript, quantitative comparisons are therefore derived from the MSM at the level of these metastable states, rather than from unweighted counts of concatenated AB-MD frames.

Accordingly, we have revised the sections “E219K and Y221A mutations facilitate proton transfer” and “Substitutions enlarge the active-site pocket to accommodate bulkier R1 and R2 groups of β-lactams”, and we have added new figures in Figure 6 and its figure supplement. The adjustments to the quantitative analyses do not affect our original conclusions.

The following is the authors’ response to the original reviews.

**Reviewer #1** (**Public review)**:Summary:This manuscript uses adaptive sampling simulations to understand the impact of mutations on the specificity of the enzyme PDC-3 β-lactamase. The authors argue that mutations in the Ω-loop can expand the active site to accommodate larger substrates.Strengths:The authors simulate an array of variants and perform numerous analyses to support their conclusions. The use of constant pH simulations to connect structural differences with likely functional outcomes is a strength.Weaknesses:I would like to have seen more error bars on quantities reported (e.g., % populations reported in the text and Table 1).

We appreciate this point. Here, the population we analyze is intended to showcase conformational differences across variants rather than to estimate equilibrium occupancies. Although each system includes 100 trajectories, they were generated using an adaptive-bandit protocol. The protocol deliberately guides towards underexplored basins, therefore conformational heterogeneity betweentrajectories is expected by design. For example, in E219K the MSM decomposition shows that in states 1, 6, and 7 the K67(NZ)–S64(OG) distance is almost entirely > 6 Å, whereas in states 2 and 3 it is almost entirely < 3.5 Å (Figure 5—figure supplement 12). These distances suggest that the hydrogen bond fraction is approximately zero in states 1, 6, and 7, and close to one in states 2 and 3. In addition, the mean first passage time of the Markov state models suggests that the formation and disruption of this hydrogen bond occur on the microsecond timescale, which is far longer than the length of each individual trajectory (300 ns). Consequently, across the 100 replicas, some trajectories exhibit very low fractions, while others display the opposite trend. Under such bimodal, protocol-induced heterogeneity, computing an error bar across trajectories mainly visualizes the protocol’s dispersion and risks being misread as thermodynamic uncertainty, which is not central to our aim of comparing conformational differences between wild-type PDC-3 and variants. We therefore do not include the error bars.

**Reviewer #2** (**Public review)**:Summary:In the manuscript entitled "Ω-Loop mutations control dynamics of the active site by modulating the 3 hydrogen-bonding network in PDC-3 4 β-lactamase", Chen and coworkers provide a computational investigation of the dynamics of the enzyme Pseudomonas-derived cephalosporinase 3 (PDC3) and some mutants associated with increased antibiotic resistance. After an initial analysis of the enzyme dynamics provided by RMSD/RMSF, the author concludes that the mutations alter the local dynamics within the omega loop and the R2 loop. The authors show that the network of hydrogen bonds is disrupted in the mutants. Constant pH calculations showed that the mutations also change the pKa of the catalytic lysine 67, and pocket volume calculations showed that the mutations expand the catalytic pocket. Finally, time-independent component analysis (tiCA) showed different profiles for the mutant enzyme as compared to the wild type.Strengths:The scope of the manuscript is definitely relevant. Antibiotic resistance is an important problem, and, in particular, *Pseudomonas aeruginosa* resistance is associated with an increasing number of deaths. The choice of the computational methods is also something to highlight here. Although I am not familiar with Adaptive Bandit Molecular Dynamics (ABMD), the description provided in the manuscript suggests that this simulation strategy is well-suited for the problem under evaluation.Weaknesses:In the description of many of their results, the authors do not provide enough information for a deep understanding of the biochemistry/biophysics involved. Without these issues addressed, the strength of the evidence is of concern.

We thank the reviewer for pointing out the need for deeper discussion of the biochemical and biophysical implications of our results. In our manuscript, we begin by examining basic structural metrics (e.g., RMSD and RMSF) which clearly indicate that the major conformational changes occur in the Ω-loop and the R2 loop. We have now added a paragraph to describe the importance of the Ωloop and highlighted it in the revised manuscript on lines 142-166 of page 6. This observation guided our subsequent focus on these regions, as well as on the catalytic site. Our analysis revealed notable alterations in the hydrogen bonding network—especially in interactions involving the K67-S64, K67N152, K67-G220, Y150-A292, and N287-N314 pairs. These observations led us to conclude that:

(1) Mutations E219K and Y221A facilitate the proton transfer of catalytic residues. This is consistent with prior experimental data showing that these substitutions produce the most pronounced increase in sensitivity to cephalosporin antibiotics (lines 210-212 in page 8 of the revised manuscript).

(2) Substitutions enlarge the active-site pocket to accommodate bulkier R1 and R2 groups of β-lactams.This is in line with MIC measurements reported by Barnes et al. (2018), which showed that mutants with larger active-site pockets exhibit markedly greater sensitivity to cephalosporins with bulky side chains than others (lines 249-259 in pages 10).

Furthermore, we applied Markov state models (MSMs) to explore the timescales of the transitions between these different conformational states. We believe that these methodological steps support our conclusions.

**Reviewer #3** (**Public review)**:Summary:This manuscript aims to explore how mutations in the PDC-3 3 β-lactamase alter its ability to bind and catalyse reactions of antibiotic compounds. The topic is interesting, and the study uses MD simulations to provide hypotheses about how the size of the binding site is altered by mutations that change the conformation and flexibility of two loops that line the binding pocket. However, the study doesn't clearly describe the way the data is generated. While many results appear significant by eye, quantifying this and ensuring convergence would strengthen the conclusions.Strengths:The significance of the problem is clearly described, and the relationship to prior literature is discussed extensively.Weaknesses:The methods used to gain the results are not explained clearly, meaning it was hard to determine exactly how some data was obtained. The convergence and uncertainties in the data were not adequately quantified. The text is also a little long, which obscures the main findings.

We thank the reviewer for the suggestion. We respectfully ask the reviewer to specify which aspects of the data-generation methods are unclear so that we can include the necessary details in the next revision. Moreover, all statistics that are reported in the manuscript are obtained from extensive analyses of 300,000 simulation frames. The Markov state models have been validated by the ITS plots and Chapman-Kolmogorov (CK) test. The two-sample t-tests were also carried out for the volume and SASA.

**Reviewer #2 (Recommendations for the authors)**:(1) Figure 1D focus on the PDC3 catalytic site. However, the authors mentioned before that the enzyme has two domains, an alpha domain and an alpha/beta domain. The reader would benefit from a more detailed description of the enzyme, its active site, AND the location of the mutants under investigation in the figure.

We have updated Figure 1D and marked the positions of all mutations (V211A/G, G214A/R, E219A/G/K and Y221A/H), which have now been highlighted as spheres.

(2) Since in the journal format, the results come before the methods. It would be interesting to add a brief description of where the results came from. For example, in the first section of the results, the authors describe the flexibility of the omega loop and the R2 loop. However, the reader won't know what kind of simulation was used and for how long, for example. A sentence would add the required context for a deeper understanding here.

At the beginning of the Results and Discussion section we now state: “To investigate how the mutations in the Ω-loop affect PDC-3 dynamics, adaptive-bandit molecular dynamics (AB-MD) simulations were carried out for each system. 100 trajectories of 300 ns each (totaling 30 μs per system) were run.”

(3) Still in the same section, the authors don't define what change in RMSF is considered significant. For example, I can't see a relevant change in the RMSF for the omega loop between the et enzyme and the E219 mutants in Figure 2D. A more objective definition would be of benefit here.

Our analysis reveals that while the wild-type PDC-3 and the G214A, G214R, E214G, and Y221A variants exhibit an average per-residue RMSF of around 4 Å in the Ω-loop, the V211A and V211G variants show markedly lower values (around 1.5 Å), and the E219K and Y221H variants exhibit intermediate values between 2 and 2.5 Å. In addition, the fluctuations around the binding site should be seen collectively along with the fluctuations in the R2-loop. Importantly, we urge the reviewer to focus on the MDLovofit analysis in Figure 2C, where the dynamic differences between the core and the fluctuating loops is clearly evident.

(4) In line 138, the authors state that "Therefore, the flexibility of these proteins is mainly caused by the fluctuations in the Ω-loops and R2-loop". This is quite a bold statement to be drawn at this point. First of all, there is no mention of it in the manuscript, but is there any domain movement? Figure 2C clearly shows that there is some mobility in omega and R2 loops. But there is no evidence shown in the manuscript that shows that "the flexibility of these proteins is mainly caused by the fluctuations in the" loops. Please consider rephrasing this sentence or adding more data, if available.

We have revised the wording to take the reviewer’s concern into account. The sentence now states: “Therefore, flexibility of PDC-3 is predominantly localized to the Ω- and R2-loops, whereas the remainder of the structure is comparatively rigid.” To further explain to the reviewer, the β lactamase enzymes are fairly rigid structures, where no large-scale domain motions occur. Instead, the enzyme communicates structurally via cross correlation of loop dynamics (https://doi.org/10.7554/eLife.66567).

(5) I guess, the most relevant question for the scope of the paper is not answered in this section. The authors show that the mobility of the omega- and R2-loops is altered by some mutations. Why is that? I wish I could see a figure showing where the mutations are and where the loops are. This question will come back in other sections.

We have updated Figure 1D to mark the positions of all mutations (V211A/G, G214A/R, E219A/G/K and Y221A/H) as spheres. The Ω- and R2-loops are also highlighted. All mutations map to the Ω-loop, indicating that these substitutions directly perturb this region. Notably, K67 forms a hydrogen bond with the backbone of G220 within the Ω-loop and another with the phenolic hydroxyl of Y150. Y150, in turn, hydrogen-bonds with A292 in the R2 loop. Together, the residue interaction network (G220– K67–Y150–A292) suggest a pathway by which Ω-loop mutations propagate their effects to the R2 loop.

(6) The authors then analyze the network of polar residues in the active site and the hydrogen bonds observed there. For the K67-N152 hydrogen bond, for example, there is a reduction in the occupancy from ~70% in the wild-type enzyme to ~30% and 40% in the mutants E219K and Y221, respectively. This finding is interesting. The question that remains is "why is that"? From the structural point of view, how does the replacement of E219 with a Lysine alter the hydrogen bond formation between K67 and N152? Is it due to direct competition? Solvent rearrangement? The reader is left without a clue in this section. Also, Figure 3B won't help the reader, since the mutated residues are not shown there. Please consider adding some information about why the authors believe that the mutations are disrupting the active site hydrogen bond network and showing it in Figure 3B.

We appreciate the comment and have updated Figures 1D and 3B to highlight the mutation sites. The change from ~70% in the wild type to ~30–40% in the E219K and Y221T variants reported in Table 1 refers to the S64–K67 hydrogen bond. In the wild type, K67 forms an additional hydrogen bond with G220 on the Ω-loop, which helps anchor the K67 side chain in a geometry that favors the S64–K67 interaction. In the variants, the mutations reshape the Ω-loop and frequently disrupt the K67–G220 contact. The loss of this local anchor increases the conformational dispersion of K67, which is consistent with the observed reduction of the S64–K67 occupancy. Furthermore, our observation that the mutations are disrupting the active-site hydrogen-bond network is a data-driven conclusion rather than a subjective inference. Across ten systems, our AB-MD simulations provided 30 µs of sampling per system. Saving one frame every nanosecond yielded 30,000 conformations per system and 300,000 in total. All hydrogen-bond and salt-bridge statistics were computed over this full ensemble. Thus, the conclusion that the mutations disrupt the active-site hydrogen-bond network follows directly from these ensemble statistics.

(7) The pKa calculations and the pocket volume calculations show that the mutations expand the volume of the catalytic site and alter the microenvironment. Is there any change in the solvation associated with these changes? If the volume expands and the environment becomes more acidic, are there more water molecules in the mutants as compared to the wt enzyme? If so, can changes in solvation be associated with the changes in the hydrogen bond network? Would a simulation in the presence of a substrate be meaningful here? (I guess it would!).

Regarding solvation, we observe a modest increase in transient water occupancy associated with the increase in volume of the pocket. The conserved deacylation water molecule is the most important and is always present throughout the simulation. Additional waters enter and leave the pocket but do not form persistent interactions that measurably perturb the hydrogen-bond network of the Ω- and R2-loops. We agree that simulations with a bound substrate would be informative. However, our study focuses on how Ω-loop mutations modulate the active site of apo PDC-3 and its variants. Within this scope, we find: (i) Amino acid substitutions change the flexibility of Ω-loops and R2-loops; (ii) E219K and Y221A mutations facilitate the proton transfer; (iii) Substitutions enlarge the active-site pocket to accommodate bulkier R1 and R2 groups of β-lactams.

(8) I have some concerns regarding the Markov State Modeling as shown here. After a time-independent component analysis, the authors show the projections on the components, which is different between wild wild-type enzyme and the mutants, and draw some conclusions from these changes. For example, the authors state that "From the metastable state results, we observe that E219K adopts a highly stable conformation in which all the tridentate hydrogen-bonding interactions (K67(NZ)-S64(OG), K67(NZ)N152(OD1) and K67(NZ)-G220(O)) mentioned above are broken". This is conclusion is very difficult to draw from Figure 5 alone. Unless the macrostates observed in the MSM can be shown (their structures) and could confirm the broken interactions, I really don't believe that the reader can come to the same conclusion as drawn by the authors here. I would recommend the authors to map the macrostates back to the coordinates and show them (what structure corresponds to what macrostate). After showing that, it makes sense to discuss what macrostate is being favored by what mutation. Taking conclusions from tiCA projections only is not recommended. I very strongly suggest that the authors revisit this entire section, adding more context so that the reader can draw conclusions from the data that is shown.

We appreciate the reviewer’s concern. In the Markov state modeling section, our objective is to quantify the timescales (via mean first passage times) associated with the formation and disruption of the critical hydrogen bonds (K67(NZ)-S64(OG), K67(NZ)-N152(OD1), K67(NZ)-G220(O), Y150(N)A292(O), N287(ND2)-N314(OD1)) mentioned above. Representative structures illustrating these interactions are shown in Figures 3B and 4A. We agree that the main Figure 5 alone does not convey structural information. Accordingly, we provide Figure 5—figure supplements 12–16. Together, Figure 5B and Figure 5—figure supplements 12–16 map structures to metastable states, whereas Figures 3B and 4A supply atomistic detail of the interactions. Author response image 1 presents selected subplots from Figure 5— figure supplements 12–14. Together with the free-energy landscape in Figure 5A, these data indicate that E219K adopts a highly stable conformation in which all three K67-centered hydrogen bonds (K67(NZ)–S64(OG), K67(NZ)–N152(OD1), and K67(NZ)–G220(O)) are broken.

**Author response image 1. sa3fig1:** TICA plot illustrates the distribution of E219K with the colour indicating the K67(NZ)-S64(OG), K67(NZ)-N152(OD1) and K67(NZ)-G220(O) distance.

(9) As a very minor issue, there are a few typos in the manuscript text. The authors might want to take some time to revisit their entire text. Examples in lines 70, 197, etc.

Thank you for your comment. We have corrected these typos.

**Reviewer #3 (Recommendations for the authors)**:This manuscript aims to explore how mutations in the PDC-3 3 β-lactamase alter its ability to bind and catalyse reactions of antibiotic compounds. The topic is interesting, and the study uses MD simulations to provide hypotheses about how the size of the binding site is altered by mutations that change the conformation and flexibility of two loops that line the binding pocket.However, the study doesn't clearly describe the way the data is generated and potentially lacks statistical rigour, which makes it uncertain if the key results are significant. As such, it is difficult to judge if the conclusions made are supported by data.

All necessary data-acquisition methods are described in the Methods section. The Markov state models have been validated by the ITS plot and the Chapman-Kolmogorov (CK) test (Figure 5—figure supplement 2–11) . The two-sample t-tests were also carried out for the volume and SASA (Table 2).

The results section jumps straight to reporting RMSD and RMSF values; however, it is not clear what simulations are used to generate this information. Indeed, the main text does not mention the simulations themselves at all. The methods section mentions that 10 independent MD simulations were set up for each system, but no information is given as to how long these were run or the equilibration protocol used. Then it says that AB-MD simulations were run, but it is not clear what starting coordinates were used for this or how the 10 replicates were fed into these simulations. Most importantly, are the RMSD and RMSF calculations and later distance distribution information derived from the equilibrium MD runs or from the AB-MD simulations?

Thank you for pointing this out. We have added “To investigate how the mutations in the Ω-loop affect PDC-3 dynamics, adaptive-bandit molecular dynamics (AB-MD) simulations were carried out for each system. 100 trajectories of 300 ns each (totaling 30 μs per system) were run.” to the Results and Discussion section. We didn’t run 10 independent MD simulations per system. We regret the typo in the Methods section that confused the reviewer. The sentence should have read – ‘All-atom MD simulations of wild-type PDC-3 and its variants were performed.’ Each system was equilibrated for 5 ns at 1 atmospheric pressure using Berendsen barostat. AB-MD simulations were initiated from these equilibrated structures. All analyses, apart from CpHMD, are based on the AB-MD trajectories.

If these are taken from the equilibrium simulations, then it is critical that the reproducibility and statistical significance of the simulations is established. This can be done by calculating the RMSD and RMSF values independently for each replicate and determining the error bars. From this, the significance of differences between WT and mutant simulations can be determined. Without this, I have no data to judge if the main conclusions are supported or not. If these are derived from the AB-MD simulations, then I want to know how the independent simulations were combined and reweighted to generate overall RMSD, RMSF, and distance distributions. Unless I misunderstand the approach, the individual simulations no longer sample all regions of conformational space the same relative amount you would see in a standard MD simulation - specific conformational regions are intentionally run more to enhance sampling, then the overall conformational distributions cannot be obtained from these simulations without some form of reweighting scheme. But no such scheme is described. In addition, convergence of the data is required to ensure that the RMSD, RMSF, and distances have reached stable values. It is possible that I am misunderstanding the approach here. But in that case, I hope the authors can clarify the method and provide a means of ensuring that the data presented is converged. Many of the differences are clear by eye, but it is important to know they are not random differences between simulations and rather reflect differences between them.

Thank you for raising this important point. In our AB-MD workflow, the adaptive bandit is used only for starting-structure selection (adaptive seeding). After each epoch, it chooses new starting snapshots from previously sampled conformations and launches the next runs. Each trajectory itself is standard, unbiased MD with no biasing potentials and no modification of the Hamiltonian. In other words, AB decides where we start, but does not alter the physics or sampling dynamics within an individual trajectory. In addition, our goal in this work is to compare variants under the same adaptive-bandit (AB) protocol, rather than to estimate equilibrium (Boltzmann) populations. Hence, we did not apply equilibrium reweighting to RMSD, RMSF, or distance distributions. However, MSM section provides reweighted reference results based on the MSM stationary distribution.

In the response to reviews, the authors state that the "RMSF is a statistical quantity derived from averaging the time series of atomic displacements, resulting in a fixed value without an inherent error bar." But normally we would run multiple replicates and get an error bar from the different values in each. To dismiss the request for uncertainties and error bars seems to miss the point. I strongly agree with the prior reviewer that comparisons between RMSF or other values should be accompanied by uncertainties and estimates of statistical significance.

Regarding the reviewers’ suggestion to present the data as a bar graph with error bars, we would like to note that RMSF is calculated as the time average of the fluctuations of each residue’s Cα atom over the entire simulation. As such, RMSF is a statistical quantity derived from averaging the time series of atomic displacements, resulting in a fixed value without an inherent error bar. We believe that our current presentation clearly and accurately reflects the local flexibility differences among the variants. Nearly all published studies report RMSF in this way, as indicated by the following examples:

Figure 3a in DOI: https://doi.org/10.1021/jacsau.2c00077

Figure 2 in DOI: https://doi.org/10.1021/acs.jcim.4c00089

Supplementary Fig. 1, 2, 5, 9, 12, 20, 22, 24, and 26 in DOI: https://doi.org/10.1038/s41467-022-293313

However, in response to the reviewers’ strong request, we present RMSF plots with error bars in our response letter.

**Author response image 2. sa3fig2:** The root-mean-square fluctuation (RMSF) profiles of wild-type PDC-3 and its variants. Blue lines show the mean RMSF across 100 independent MD trajectories for each system; red translucent bands denote the standard deviation across trajectories. The Ω-loop (residues G183 to S226) is highlighted in yellow, and the R2-loop (residues L280 to Q310) is highlighted in blue.

It was good to see that convergence of the constant-pH simulations was shown. While it can be challenging to get absolute pH values from the implicit solvent-based simulations, the differences between the systems are large and the trends appear significant. I was not clear how the starting coordinates were chosen for these simulations. Is the end point of the classical simulations, or is a representative snapshot chosen somehow?

To ensure comparison, all systems used the X-ray crystal structure (PDB ID: 4HEF) with T79A substitution as the initial structure. The E219K and Y221A mutants were generated in silico using the ICM mutagenesis module. We have added the clarification in Methods section: “The starting structures were identical to those used for AB-MD.”

Significant figures: Throughout the text and tables, the authors present data with more figures than are significant. 1071.81+-157.55 should be reported as 1100 +/ 160 or 1070 =- 160 . See the eLife guidelines for advice on this.

Thank you for your suggestion. We have amended these now.

The manuscript is very long for the results presented, and I feel that a clearer story would come across if the authors shortened the text so that the main conclusions and results were not lost.

We appreciate the suggestion. We examined the twenty most recent research articles published in eLife and found that they are either longer than or comparable in length to our manuscript.